# Non-Invasive Dengue Diagnostics—The Use of Saliva and Urine for Different Stages of the Illness

**DOI:** 10.3390/diagnostics11081345

**Published:** 2021-07-26

**Authors:** Mahathir Humaidi, Wei Ping Tien, Grace Yap, Choon Rong Chua, Lee Ching Ng

**Affiliations:** 1Environmental Health Institute, National Environment Agency, Singapore 138667, Singapore; Tien_Wei_Ping@nea.gov.sg (W.P.T.); Grace_YAP@nea.gov.sg (G.Y.); Rachel_CHUA@nea.gov.sg (C.R.C.); 2School of Biological Science, Nanyang Technological University, Singapore 639798, Singapore

**Keywords:** dengue, diagnostics, non-invasive, urine, saliva

## Abstract

Dengue diagnosis is largely dependent on clinical symptoms and routinely confirmed with laboratory detection of dengue virus in patient serum samples collected via phlebotomy. This presents a challenge to patients not amenable to venipuncture. Non-invasive methods of dengue diagnosis have the potential to enhance the current dengue detection algorithm. In this study, samples from dengue infected patients were collected between January 2012 until September 2012 and September 2013 until December 2013 in two different setups. Panel A samples (blood, urine, and saliva) were collected daily when the 39 patients were hospitalised and during their follow-up visits while Panel B samples (saliva) were collected from 23 patients during the acute stage of dengue. Using DENV PCR on Panel A, from day 2 to day 4 post fever onset, serum showed the best overall positivity followed by saliva and urine (100%/82.1%/67.9%). From day 5 until day 10 post fever onset, serum and urine had similar positivity (67.4%/61.2%), followed by saliva (51.3%). Beyond day 10 post fever onset, DENV was undetectable in sera, but urine and saliva showed 56.8% and 28.6% positivity, respectively. DENV in urine was detectable up until 32 days post fever. Panel B results showed overall sensitivity of 32.4%/36% (RNA/NS1) for DENV detection in saliva. Our results suggest that the urine-based detection method is useful especially for late dengue detection, where DENV is undetected in sera but still detectable in urine. This provides a potential tool for the physician to pick up new cases in an area where there is ongoing dengue transmission and subsequently prompt for intensified vector control activities.

## 1. Introduction

Dengue is the world’s most common mosquito-borne virus, with approximately three billion people at risk of infection each year. Across approximately 100 countries, an estimated 50 million infections per year challenge the public health infrastructure of the affected countries. An accurate and quick laboratory diagnosis is essential for the clinician to distinguish initial-phase dengue from other diseases with similar clinical symptoms. This allows for the efficient allocation of limited resources for vector control activities [1,2,3,4,5,6,7,8]. There are two methods in the laboratory diagnosis of dengue. The direct method involves the detection of isolated DENV, viral antigens such as NS1 protein, or genomic sequence via reverse transcriptase-polymerase chain reaction (RT-PCR). The indirect method requires the detection of high levels of dengue-specific IgM, IgG, or IgA antibodies. Currently, RT-PCR assay or NS1 detection by means of enzyme-linked immunosorbent assay (ELISA) or the lateral-flow rapid test is sufficient for the confirmation of acute dengue infection. Serologic assays of dengue-reactive IgM and IgG antibodies can be used to show if a patient has a primary or a secondary dengue infection. ELISA-based IgM kits when compared to rapid test IgM kits have higher sensitivities (61.5–99.0% vs. 20.5–97.7%) and higher specificities (79.9–97.8% vs. 76.6–90.6%) [9]. It must also be noted that dengue IgM cross reacts with all four DENV serotypes and other flaviviruses such as West Nile virus (WNV), St. Louis encephalitis virus (SLE), Japanese encephalitis virus (JEV) and Yellow fever virus (YFV) [9,10]. IgG is not suitable for early diagnosis of primary infection cases as it can be detected only after 10 days from illness onset. In secondary infection, however, rapid increase of IgG can be detected from as early as day 4 after illness onset [11]. Dengue IgA has been detected in serum of febrile patients between days 8 and 11 after fever onset of a primary dengue infection. During secondary infection, IgA increases almost immediately after fever onset [12,13].

Employing a saliva-based technique in dengue diagnosis provides an easy and non-invasive method for sample collection. Poloni et al. and Mizuno et al. were also able to detect dengue virus genome in saliva by real time RT-PCR, while Ravi Banavar et al. demonstrated 100% sensitivity and specificity with seropositive saliva samples on an IgG ELISA kit [14,15,16].

Vazquez et al. have shown that Dengue IgA and IgG but not IgM can be detected in the urine of both primary and secondary dengue infected patients [13,17]. Comparing NS1 antigen positive rates, Saito et al. detected similar rates in urine and serum on days 11 to >15 post fever onset [18]. In a study of Japanese travelers who were infected with dengue abroad, the DENV genome positivity rate for urine and serum samples was 30% and 79%, respectively, from days 1–5 after disease onset and on days 10–16 after disease onset; the corresponding positivity rate was 61%(urine) and 11%(serum) [19]. Yingsiwaphat et al. were able to isolate live DENV from the urine of both acutely infected patients and from patients 14 days post defervescence [20]. Similar findings were observed in the Common Marmoset animal model. In those studies, at 1–5 days following inoculation of DENV into an animal host, DENV genome in urine samples was detected, albeit at a lower rate than serum samples (10% vs. 94%). Following the trend of human samples at later phases of dengue (11–14 days after disease onset), DENV genome in the animal urine was presented at a higher rate compared to serum samples (27% vs. 0%). Conversely, no DENV genome was detected in urine samples from marmosets with secondary DENV infection. This animal model recapitulates clinical aspects of DENV infection and demonstrates DENV genome in urine, suggesting the potential for the use of marmosets in DENV pathogenesis studies [21].

Ma et al. have reported the first confirmation of DENV serotype-2 complete genome in the urine of a traveler. The DENV genomic material was extracted from urine collected at 8 and 18 days post disease onset. Phylogenetic analysis indicated that the sample strain was strongly located within the DENV-2 group. This breakthrough has further established the utility of urine samples in the genotyping of DENV-infected patients beyond the typical detection window of serum-based DENV antigenic and genomic assays. Such results can be used to manage the risk of DENV introduction from travel-associated DENV infections [22]. In this study, we evaluated the performances of RT-PCR and DENV NS1 in saliva and urine samples collected from serum-based confirmed dengue patients in Singapore.

## 2. Materials and Methods

### 2.1. Study Population

Panel A: Blood, saliva and urine samples were collected from symptomatic patients seen at Tan Tock Seng Hospital Dengue Clinic, setup under the Singapore Stop Dengue programme, which ran from January 2012 until September 2012. All samples were transported to EHI on ice, processed immediately and kept at −80 °C until testing. Real-time reverse transcriptase-polymerase chain reaction (RT-PCR) was performed for diagnosis and positive samples were serotyped using FRET probe-based multiplex RT-PCR assay (Roche Diagnostics, Basel, Switzerland) [23]. Once the sample was positive for dengue virus, daily samples were collected from patients at the Dengue Clinic until they were discharged. Convalescence samples were collected three weeks later during a follow-up visit. Samples from 39 dengue-positive patients were collected for this study. Collections ranged from two days post fever until 41 days post fever. Twenty-four dengue-negative samples were used to calculate specificity.

Panel B: Saliva samples from 23 patients were collected from September to December 2013. They were positive for DENV based on the dengue NS1 Ag rapid test kit (SD Bioline, Standard Diagnostics, Inc., Davis, CA, USA) at Tan Tock Seng Hospital Dengue Clinic. All samples were collected during acute phase within 7 days of the febrile stage and were immediately processed with residuals stored in −80 °C.

### 2.2. Sample Collection

Saliva was collected using the Oracol saliva collection swab (Malvern Medical Development Ltd., Worcester, UK). In order to stimulate active saliva secretion, patients were instructed to scrub their inner upper and lower cheeks 10 times on both sides and place the swab under their tongue for 1 min [24]. For urine collection, patients were instructed to collect their mid-stream urine.

### 2.3. DENV Genome Detection

#### 2.3.1. Semi-Nested Conventional RT-PCR

DENV genome in all saliva and urine samples were examined by semi-nested conventional RT-PCR adapted from Lanciotti et al. [25]. Viral RNA was extracted from 140 µL of urine or saliva sample using the QIAmp Viral RNA Mini Kit (Qiagen, Hilden, Germany) according to manufacturer’s instructions. Five microliters of extracted viral RNA was added to the Access Quick RT-PCR System (Promega Corporation, Madison, WI, USA) with 12.5 pmol of the consensus-round primers. The 50 uL RT-PCR reaction mix was amplified in a Veriti 96-Well Fast Thermal Cycler (Applied Biosystems Inc, Waltham, MA, USA). The PCR program for this consensus-round consisted of a 30 min RT step at 45 °C, followed by a 2 min initial denaturation step at 94 °C and 40 cycles of PCR steps (94 °C for 30 s, 55 °C for 30 s and 72 °C for 1 min), ending with a 5 min final extension step at 72 °C.

Two microliters of the RT-PCR product was then mixed with Phusion Flash Hi-Fidelity PCR Master Mix (Thermo Fisher Scientific Inc, Waltham, MA, USA) with 10 pmol of D1 consensus-round forward primer and 5 pmol each of the specific-round reverse primers. The 20 µL PCR reaction mix was amplified in the same PCR machine. The PCR program for the specific round consisted of a 10 s initial denaturation step at 98 °C and 30 cycles of PCR steps (98 °C for 1 s, 59 °C for 5 s and 72 °C for 8 s), ending with a 1 min final extension step at 72 °C.

The PCR products were run in an electrophoresis gel with 1.5% agarose and GelRed TM Nucleic Acid Gel Stain (Biotium Inc, San Francisco, CA, USA). Visualisation and gel documentation were carried out in a Vilber Lourmat UV-Transilluminator with a Kodak CCD digital camera (Vilber Lourmat, Collegien, France).

To determine the performance of this semi-nested PCR on saliva and urine compared to serum, the fluid samples from PCR-negative patients were spiked and serial diluted separately with 4 serotypes of laboratory-cultured DENV up to 10^−3^ plaque forming units (PFU)/mL. Viral RNA was extracted from diluted spiked samples with QIAmp Viral RNA Mini Kit (Qiagen). Semi-nested conventional RT-PCR was carried out, as previously described.

#### 2.3.2. DENV NS1 Antigen Tests

Presence of DENV NS1 antigen in samples was tested using Biorad NS1 rapid test (Bio-Rad Laboratories, Marnes-la-Coquette, France) and Platelia dengue NS1 Ag test (Bio-Rad Laboratories). The rapid test kit was used in Panel A for its speed of detection but NS1 antigen was not detected from the saliva samples. Thereafter, the more sensitive ELISA assay was utilised in Panel B [26]. Saliva samples from dengue-positive patients in Panel B were subjected to the semi-nested DENV PCR and Bio-Rad Platelia dengue NS1 Ag test for the detection of DENV RNA and NS1 antigens. The entire assay preparation process, running, and the raw data analysis were done according to the manufacturer’s instruction.

#### 2.3.3. Virus Isolation

PCR positive saliva and urine samples were subjected to virus isolation using *Aedes albopictus* cells C6/36 (ATCC CRL-1660) and Vero cells (ATCC CCL-81). 

#### 2.3.4. Statistical Analysis

The proportions of positive tests were calculated by dividing the number of positive tests by the total number of tests in the sample. To estimate uncertainty over obtained proportions, 95% confidence intervals (CIs) were calculated. Statistically significant differences were assumed when the 95% CIs of 2 samples did not overlap. Non-significant differences are indicated with the legend [N.S.]. Microsoft Excel (2021) was used for all statistical analyses.

## 3. Results

### 3.1. Panel A

#### 3.1.1. Sensitivity of Semi-Nested DENV PCR Using DENV-Positive Sera, Urine, and Saliva

Within the first four days of fever onset, serum samples showed 100% positivity on day 1 to 3 while urine and saliva samples showed peak positivity only on day 4 (Figure 1 and Figure 2). From day 5, positivity rates decreased for sera, urine, and saliva. Beyond day 10, DENV was not detected in sera but remained detectable in urine until day 32. On day six, urine showed higher positivity than serum (60% vs. 58.3%) [N.S.] and on day eight, both urine and saliva had higher positivity than serum (63.2% and 44.4% vs. 36.4%) [N.S.]. DENV detection in urine also remained positive up to day 13, and well into convalescence with overall positivity of 53.1% (day 23 to day 32). All serotypes detected in urine and saliva matched those detected in the sera.

#### 3.1.2. Sensitivity of DENV NS1 Using DENV-Positive Sera, Urine, and Saliva

When comparing sensitivity of NS1 detection in urine versus sera, the latter achieved 100% positivity from day 2 until day 10 with a slight drop to 90% on day 4 (Figure 3). NS1 in urine was only detected on the fourth day of fever onset at 44.4% positivity and increased to 65.5% on the fifth day and progressively decreased after the sixth day onwards. As the disease progressed, sensitivity dropped to 28.6% on day 10. Similar to that in sera, sensitivity of urine followed a similar trend over time. No DENV NS1 could be detected in saliva samples when using a Biorad NS1 rapid test.

#### 3.1.3. Virus Isolation

DENV could be isolated from 73.3% (*n* = 22) of the PCR-positive saliva samples. None could be isolated from urine samples. 

### 3.2. Panel B

The aim of this panel was to test if the presence of NS1 antigens in saliva could be detected via NS1 antigen ELISA, which is more sensitive than the rapid test kit utilised in Panel A. Unlike Panel A, the swabs from Panel B were processed immediately after saliva collection.

We found the positivity rates for (RNA/NS1) decreased from 100%/100% to 69.2%/46.2% for the first three days of fever onset (Table 1). Subsequently, there was no consistent detection of DENV RNA and NS1 (Figure 4).

## 4. Discussion

Our results demonstrated the efficacy of urine and saliva samples for both DENV genomic detection and DENV NS1. Anders et al. (2012) reported that a using NS1-ELISA kit gave 64.7% detection of NS1 from patients with less than or equal to 72 h of illness, but there was no DENV RNA detection from oral swab samples [27]. Andries et al. (2015) reported overall sensitivity of 76.4%/31.6% for DENV RNA/NS1 in saliva of dengue-infected patients from the first three days after fever onset [28]. Similarly, in our Panel B saliva samples, we observed 69.2%/46.2% detection for DENV RNA/NS1 in saliva (up to 3 days after fever onset). Anders et al. used a similar swab-based collection method for saliva; however, they stored the saliva in PBS instead of UTM after sample collection. Andries et al. used the direct spitting method for collection and VTM for storage. These differences of transport media (PBS vs. UTM/VTM) might have played a role in preserving the integrity of DENV in saliva before the extraction of RNA.

In Panel B studies, we showed 32.4%/36% for the overall positivity of DENV RNA/NS1 in saliva (Figure 4). Unlike in Panel A, we observed NS1 detection in Panel B, as NS1 ELISA is more sensitive in detecting NS1 antigen in saliva than the rapid test kit [29,30,31,32]. We want to highlight that the saliva samples in Panel B were freshly collected from the patients and processed immediately, while Panel A samples had to be transported and then stored in −80 °C before being processed. The difference between the kits’ sensitivity and the improved sample handling might have contributed to the difference in performance between our two assay panels.

To date, other groups have reported the diagnostic performance of DENV RNA and NS1 in both saliva and urine samples. Korhonen et al. (2014) showed an overall sensitivity of 60%/56% for RNA/NS1 in saliva, while urine samples had overall sensitivity of 64%/54% for RNA/NS1 [33]. Andries et al. (2015) demonstrated an overall sensitivity of 41.6%/14.5% for RNA/NS1 in urine samples. Hirayama et al. (2012), who worked on urine samples, observed that the detection of the DENV genome in dengue-positive urine samples surpassed that of serum (78% vs. 33%) at day 8 of fever onset [19]. Chuansumrit et al. (2011) presented the first evidence of NS1 detection (68.4%) in the urine samples from DENV-positive patients [34].

In this study, we investigated the detection rate of RNA and NS1 in sera, saliva, and urine for dengue diagnosis from samples collected over a long span of 41 days. Similar to studies conducted elsewhere, our results showed a similar trend where the positivity of the detection rate in urine surpassed that of sera at day 6 of fever onset (60% vs. 58.3%) [N.S.]. We also detected an overall positivity of 63.2% for urine vs. 80% for sera in the first 7 days of fever onset, indicating that urine samples can be used for acute diagnosis. We found that if a patient presents themself later in the disease phase, DENV RNA could be detected in urine from day 11 until day 32 (during convalescence stage), when DENV genomic material is no longer detectable in sera. Other studies such as that of Mizuno et al. (2007) detected DENV RNA in urine and saliva samples up to 14 days after onset of fever [15]. We have also showed that the window period where NS1 can be detected in urine samples is longer than that of sera (Figure 3). Hence, urine samples can be an alternative to blood samples when a patient presents themself to the physician post-fever onset from 6 to 10 days [32].

In our studies, we attempted to isolate the virus from saliva and urine samples using two different cell lines (C6/36 and Vero cells). Korhonen et al. were previously unsuccessful in isolating DENV virus from saliva; however, we achieved an isolation rate of 73.3% from our PCR-positive saliva samples. None of the attempts were successful for urine samples; this is similar to previous reports [19,33]. 

In the acute phase of infection (first three days), DENV RNA can be detected with up to 87.5% sensitivity in saliva samples. However, the NS1 performance in saliva samples is unsatisfactory at 46.2% sensitivity. Urine NS1 ELISA showed much better detection than saliva NS1 ELISA with 61.0% vs. 36.0% overall positivity during the acute phase (up to 7d fever onset). Urine NS1 strip also produced better detection than saliva NS1 ELISA with 51.4% vs. 36% positivity. Banavar et al. demonstrated 100% positivity of DENV IgG using an IgG ELISA kit on seropositive saliva samples [16]. In the endemic field setting, an NS1-Rapid test kit is preferred as it is cheap, fast, easy to use and does not require a dedicated laboratory. Current tools need to be further developed and improved for the rapid Dengue-NS1 test kits to be used as a saliva-based diagnostic test. 

The presence of DENV in urine of recovered patients has yet to be elucidated, but perhaps could be explained using the WNV model where chronic renal failure is observed in those who have recovered from the disease [35,36,37,38]. Bandeira et al. postulated that an inflammatory infiltration with macrophages in the genital tract may be a source of CHIKV [39]. DENV and YFV virus RNA were also recently reported in semen collected from respective DENV and YFV patients. Semen along with urine are promising non-invasive clinical specimen for the diagnosis of patients with neglected diseases [39,40]. Chia et al. described five ZIKV-DENV coinfections out of a total of 163 cases of ZIKV infection during a ZIKV outbreak in Singapore [41]. As arboviral diseases are increasingly presented as co-infections, it is pertinent that non-serological and non-invasive tests are developed and optimized further to prevent misdiagnosis and false-negatives in convalescent patients [42].

## Figures and Tables

**Figure 1 diagnostics-11-01345-f001:**
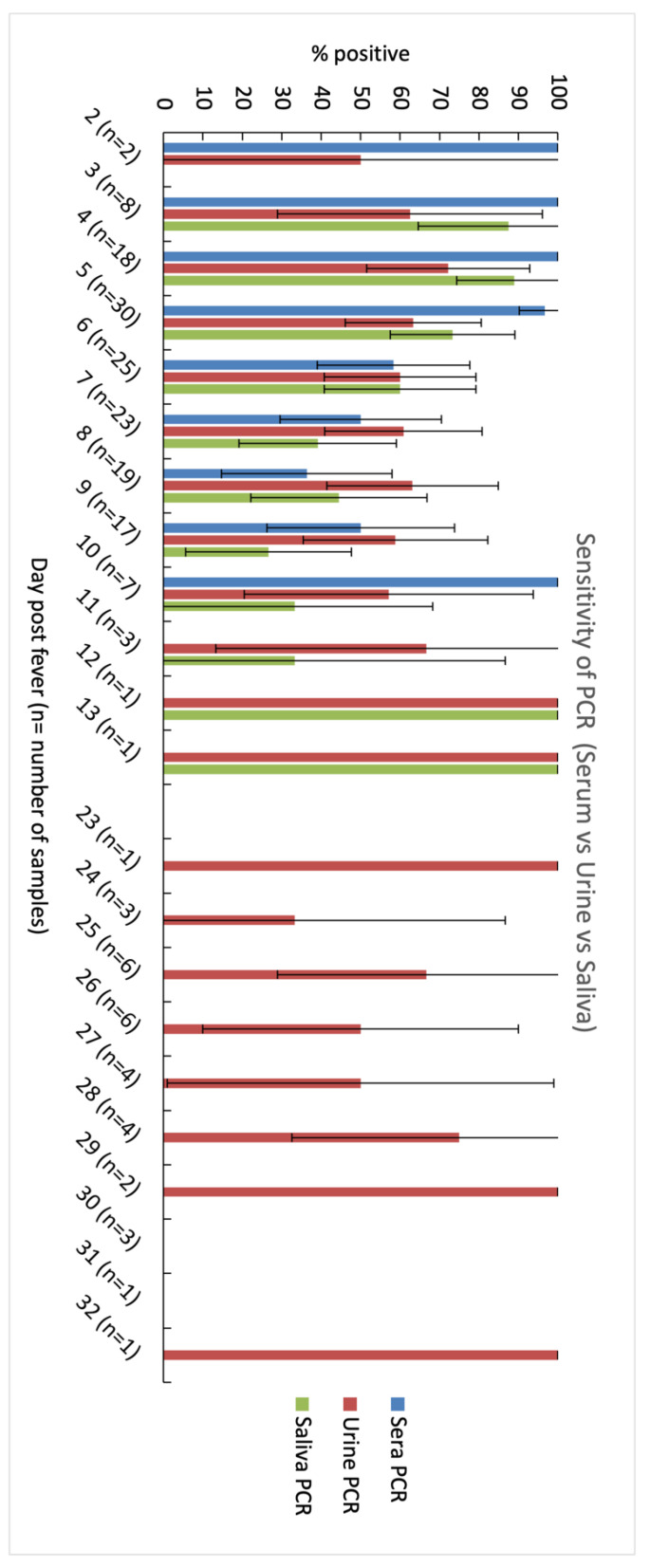
Comparative daily analysis of dengue viral RNA in serum, urine and saliva samples using semi-nested DENV PCR. DENV RNA was detected in urine samples up to 32 days of fever, compared with up to 30 days and 10 days in saliva and serum samples, respectively. *n* is the number of patients in each day. The whiskers represent 95% CI for each sample.

**Figure 2 diagnostics-11-01345-f002:**
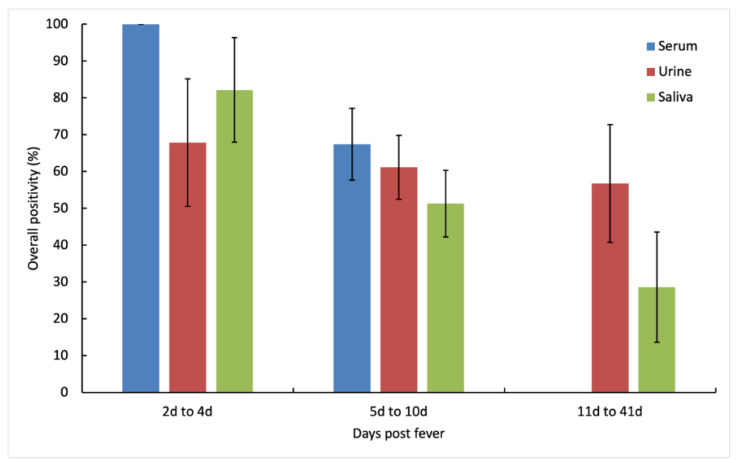
Overall positivity of DENV RNA in serum, urine, and saliva samples. From 2 days to 4 days after fever onset, serum showed the best overall positivity followed by saliva and urine, whereas from 5 days to 10 days after fever onset, urine showed better overall positivity than saliva. Urine was also the best sample for detecting DENV RNA from 11 days to 41 days after fever onset. The whiskers represent 95% CI for each sample.

**Figure 3 diagnostics-11-01345-f003:**
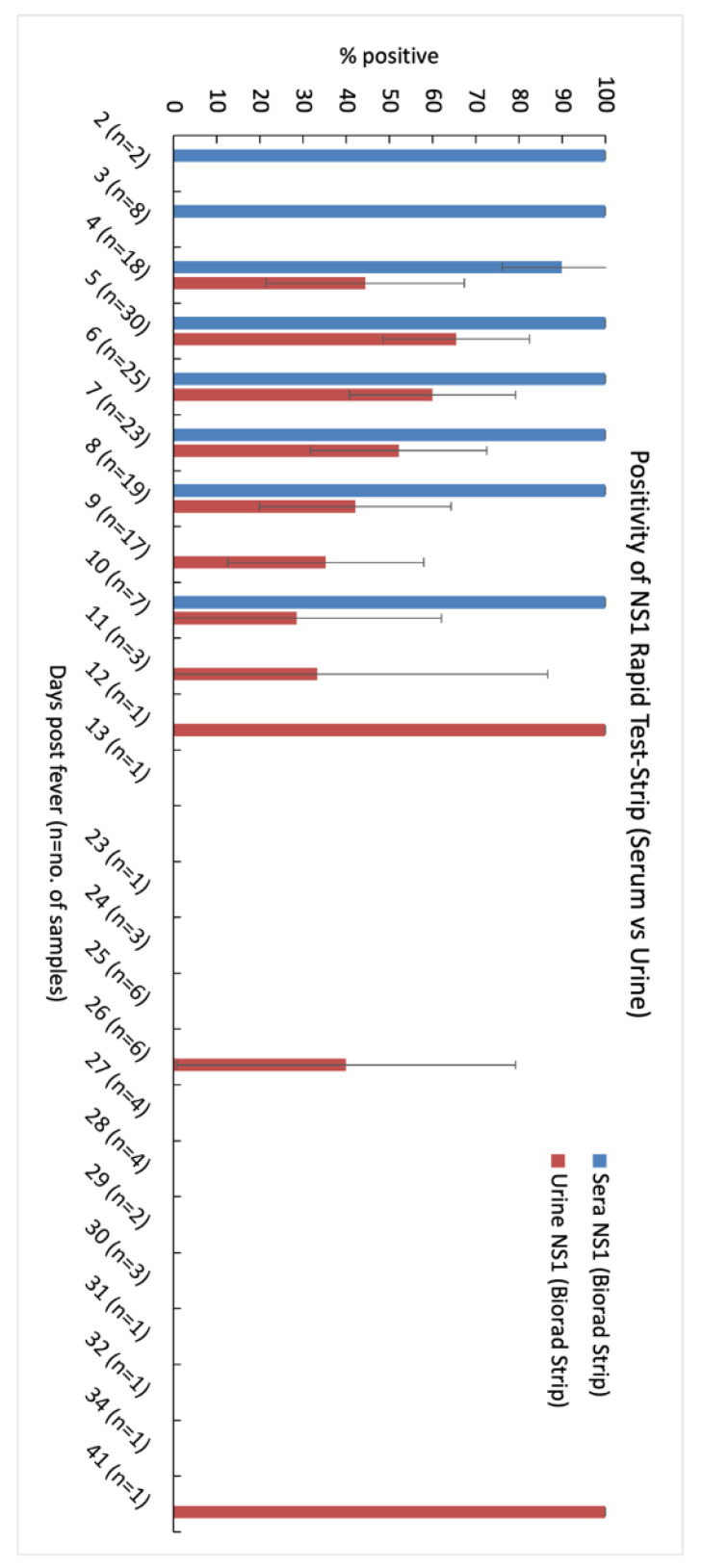
Comparative analysis of detection of dengue NS1 antigens in serum and urine samples using a Bio-Rad NS1 rapid test kit. DENV NS1 was detected in urine samples up to 12 days of fever, and up to 10 days in acute serum samples. In convalescent samples (23d to 41d), DENV NS1 was detected in 3 out of 34 samples. There was no detection of DENV NS1 in saliva samples. *n* is the number of patients in each day. The whiskers represent 95% CI for each sample.

**Figure 4 diagnostics-11-01345-f004:**
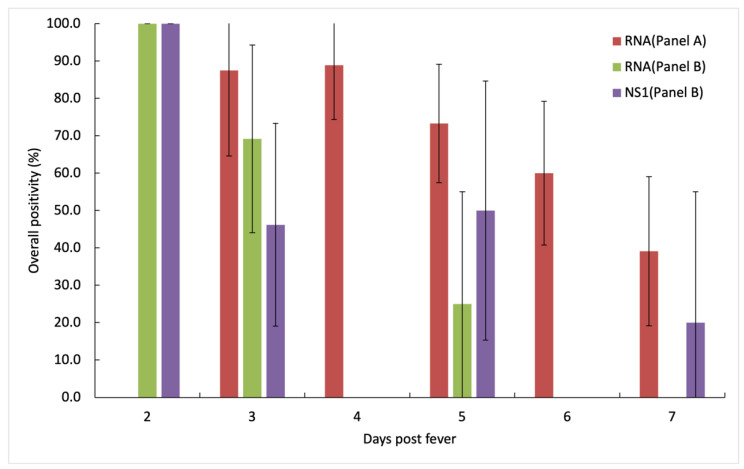
Comparative analysis of detection of dengue viral RNA and NS1 antigens in saliva samples collected in Panel B. Similar to Panel A, positivity of DENV RNA showed a downward trend as the disease progressed and diminished from day 6 onwards. DENV NS1 positivity in saliva also showed a downward trend but it remained detectable at day 7 after fever onset. The whiskers represent 95% CI for each sample. Overall positivity of Panel B (RNA/NS1) is 34.2%/36%.

**Table 1 diagnostics-11-01345-t001:** Comparative analysis of positivity of DENV RNA from Panels A and B, and NS1 in saliva collected from Panel B patients according to the days of fever.

Days of Fever	Positivity (%)		
	RNA (Panel A)	RNA (Panel B)	NS1(Panel B)
2	0 (*n* = 2)	100 (*n* = 1)	100 (*n* = 1)
3	87.5 (*n* = 8)	69.2 (*n* = 13)	46.2 (*n* = 13)
4	88.9 (*n* = 18)	0 (*n* = 4)	0 (*n* = 4)
5	73.3 (*n* = 30)	25 (*n* = 8)	50 (*n* = 8)
6	60 (*n* = 25)	0 (*n* = 2)	0 (*n* = 2)
7	39.1 (*n* = 23)	0 (*n* = 5)	20 (*n* = 5)

## Data Availability

The data presented in this study are available on request from the corresponding author. The data are not publicly available due to data security reasons.

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
