# Peer review of "Non-Invasive Dengue Diagnostics—The Use of Saliva and Urine for Different Stages of the Illness"

_diagnostics, 2021, doi:10.3390/diagnostics11081345_

Round 1
Reviewer 1 Report
In this manuscript, the authors report non-invasive Dengue diagnostics using the saliva and urine to detect dengue. Even though it is an important data set regarding virus detection and time sensitivity that can help diagnosis, so it deserves to be published, the manuscript is too raw in the current presentation and must be improved before publication. I recommend a major revision and profound changes before publication. Please, consider my points:
ABSTRACT
Well-written and clear.
INTRODUCTION
This section is well-written and sufficient background is available. Minor issues:
Line 52. A period is missing.
Lines 55 and 56 - Please, fix et. al. (et al. ). There are more occurrences throughout the manuscript. It is not standardized. Please, standardize it.
METHODS:
The methods section looks fine. However, there is a major issue:
Regarding the ethical aspects, IRB approval, and consent of the biological samples – (even if it was unidentified after collection, the authors should add a detailed section about all the procedures).
Another issue: Statistical analysis – methods? There is no section about it.
Minor issues:
The authors have used Conventional RT-PCR to analyze the samples and validate the study. The best method would be qRT-PCR for sensitivity and reliability. The authors should explain in detail the reason why this was the option over qRT-PCR (costs, accessibility to laboratories, etc) and explain the limitations.
Line 91 – Space after period
Line 92. Remove 1 period
Line 94 – Space before: (Roche)
RESULTS:
Major issues: Figures are unfinished and totally raw from excel. The error bars are out of the limits and there are no descriptions of the errors (SEM or SD). There is no statistical analysis on the bars to show significant differences, nor methods describing the stats (also on the legends – for the figures ). The authors should complete the legends, improve the figures’ presentation, and include statistical tests on the figures and methods.
The legend “days of fever”, is it better “days after fever”. Or “days post fever”? What is the authors' opinion about it?
Line 162 – “positiveup” , missing a space
Line 165-167: Is this part of the article:
“This section may be divided by subheadings. It should provide a concise and precise description of the experimental results, their interpretation, as well as the experimental conclusions that can be drawn.”
Author Response
Response to Reviewer 1 Comments
Point 1 :Line 52. A period is missing. Lines 55 and 56 - Please, fix et. al. (et al. ). There are more occurrences throughout the manuscript. It is not standardized. Please, standardize it.
Response 1: We will address these minor points in the Introduction as advised.
Point 2: Regarding the ethical aspects, IRB approval, and consent of the biological samples – (even if it was unidentified after collection, the authors should add a detailed section about all the procedures).
Response 2: We will refer to the approval documentation recorded under the National Healthcare Group Domain Specific Review Board DSRB/E/09/00432 for Panel A and NHG DSRB Ref: 2013/00111 for Panel B, and append the relevant details in the revision.
Point 3: Another issue: Statistical analysis – methods? There is no section about it.
Response 3: We will include a statistical analysis section in the Methods revision.
Point 4: The authors have used Conventional RT-PCR to analyze the samples and validate the study. The best method would be qRT-PCR for sensitivity and reliability. The authors should explain in detail the reason why this was the option over qRT-PCR (costs, accessibility to laboratories, etc) and explain the limitations.
Response 4: The conventional RT-PCR protocol is part of the routine dengue diagnostic workflow in our laboratory. Therefore, the required resources are in ready supply and furthermore, we are technically proficient with this method. We will address this point in the Methods revision.
Point 5: Line 91 – Space after period; Line 92. Remove 1 period; Line 94 – Space before: (Roche)
Response 5: We will address these minor points in the Methods as advised.
Point 6: Major issues: Figures are unfinished and totally raw from excel. The error bars are out of the limits and there are no descriptions of the errors (SEM or SD). There is no statistical analysis on the bars to show significant differences, nor methods describing the stats (also on the legends – for the figures ). The authors should complete the legends, improve the figures’ presentation, and include statistical tests on the figures and methods.
Response 6: We acknowledge these deficiencies in the Results and will address them completely in the revision.
Point 7: The legend “days of fever”, is it better “days after fever”. Or “days post fever”? What is the authors' opinion about it?
Response 7: We will standardise the legend to “days after fever” in the
Results revision.
Point 8: Line 162 – “positiveup” , missing a space
Response 8: We will address this minor point in the Results as advised.
Point 9: Line 165-167: Is this part of the article: “This section may be divided by subheadings. It should provide a concise and precise description of the experimental results, their interpretation, as well as the experimental conclusions that can be drawn.”
Response 9: This is not part of the article and will be deleted in the Results revision.
Reviewer 2 Report
The variant of dengue fever determination proposed by the authors of the article using saliva and urine is a good solution, especially in cases when it is no longer possible to establish the presence of the disease by blood plasma. There are some small remarks to the authors: 1. Drawings of very poor quality, captions are not readable. I recommend redoing. 2. In table 1, numbers are indicated in brackets, what is this? If the number of patients, then why is it either increasing or decreasing depending on the time of observation?
Reviewer 3 Report
The presented studies are certainly important from the point of view of human health. However, I have a few comments which I have listed below. Secondly, the number of patients was negligible and the research was carried out only in a relatively short series. Therefore, it is difficult to generalize the obtained results to a larger population. This requires the authors' explanations.
line 77-86, the authors described the problem well in the introduction, but I lack information about the novelty of this work. Other authors have also performed similar studies. What is the difference between the conducted research and the others?
line 165-167, please delete this, this is the text from the form for Diagnostics journal
figure 1, how to explain 100% detectability of DENV in blood at day 10 ??
figure 4, the presented results in the figure are inconclusive. How do you explain that?
line 225-226, where does this value come from: 34.2% / 36%? It does not follow from Figure 4.
lines 270-281, what is the final conclusion from the conducted research. Please present it best in the table.
Author Response
Response to Reviewer 3 Comments
Point 1 : line 77-86, the authors described the problem well in the introduction, but I lack information about the novelty of this work. Other authors have also performed similar studies. What is the difference between the conducted research and the others?
Response 1: We believe that our sampling protocol allowed us to track the DENV patients for longer span of time. We collected the patients’ daily samples throughout their hospital stay and subsequently 3 weeks later for convalescence sampling. Hence our sample collection ranged from 2 till 41 days post fever. Out of which, we were able detect DENV RNA in urine up to 32 days post fever whereby in our literature review, Ma et al. detected DENV RNA up till 18 days post fever.
Point 2: line 165-167, please delete this, this is the text from the form for Diagnostics journal
Response 2: We will delete this text in the Results revision.
Point 3: figure 1, how to explain 100% detectability of DENV in blood at day 10 ??
Response 3: Patients were being hospitalised and discharged throughout day 2 and day 13 post fever of the Panel A study. Between day 9 and day 10, we can infer that at least 10 patients had been discharged and we may assume the remaining 7 patients on day 10 of having 100% dectectability in blood. On day 9, there were at least 8 patients with DENV still detectable in blood.
Point 4: figure 4, the presented results in the figure are inconclusive. How do you explain that?
Response 4: We admit due lower sampling power in Panel B, at least, that the RNA results from Panel A are more consistent than from Panel B, but the takeaway from figure 4 shown from Panel B is the detectability of NS1 in saliva and the apparent downward trend as disease progresses.
Point 5: line 225-226, where does this value come from: 34.2% / 36%? It does not follow from Figure 4.
Response 5: It refers to the overall positivity from day 2 till day 7 post fever which is not indicated in Figure 4 or Table. We will include this value in the Results revision.
Point 6: lines 270-281, what is the final conclusion from the conducted research. Please present it best in the table.
Response 6: We will include an overall table to better illustrate the final conclusion in the revision.
Point 7: Secondly, the number of patients was negligible and the research was carried out only in a relatively short series. Therefore, it is difficult to generalize the obtained results to a larger population. This requires the authors' explanations.
Response 7: Referring to Panel A, we should ideally be able to sample the 39 patients consistently throughout the time-series. However due to the opportunistic nature of the recruitment, we had patients who entered the sampling at different time points due to them being first diagnosed at different days post fever. Subsequently, patients were both hospitalised and discharged at different days post fever. The short series of 13 days during the hospitalisation was designed based on the feature of the disease where DENV RNA in blood can generally be detected up to 10 days post fever. With regards to the recruitment numbers, this is a pilot study which shows the utility of urine sampling. We may increase the sampling population in future studies if more robust statistical power is needed.
Round 2
Reviewer 3 Report
I have no more comments. The answers are satisfactory
Author Response
We have Microsoft Word to scan for spelling and grammar mistakes, and corrected the manuscript as suggested by the language correction tool.